# Improved Estimation of Winter Wheat Aboveground Biomass Using Multiscale Textures Extracted from UAV-Based Digital Images and Hyperspectral Feature Analysis

Yuanyuan Fu [1,2,3], Guijun Yang [2,3,*], Xiaoyu Song [2,3], Zhenhong Li [2,4], Xingang Xu [2,3], Haikuan Feng [2,3] and Chunjiang Zhao [2,3]

1   College of Information and Management Science, Henan Agricultural University, Zhengzhou 450002, China; fuyy@nercita.org.cn
2   Key Laboratory of Quantitative Remote Sensing in Agriculture of Ministry of Agriculture, Beijing Research Center for Information Technology in Agriculture, Beijing 100097, China; songxy@nercita.org.cn (X.S.); Zhenhong.Li@newcastle.ac.uk (Z.L.); xuxg@nercita.org.cn (X.X.); fenghk@nercita.org.cn (H.F.); zhaocj@nercita.org.cn (C.Z.)
3   National Engineering Research Center for Information Technology in Agriculture, Beijing 100097, China
4   School of Engineering, Newcastle University, Newcastle upon Tyne NE1 7RU, UK
*   Correspondence: yanggj@nercita.org.cn; Tel.: +86-10-5150-3215

**Abstract:** Rapid and accurate crop aboveground biomass estimation is beneficial for high-throughput phenotyping and site-specific field management. This study explored the utility of high-definition digital images acquired by a low-flying unmanned aerial vehicle (UAV) and ground-based hyperspectral data for improved estimates of winter wheat biomass. To extract fine textures for characterizing the variations in winter wheat canopy structure during growing seasons, we proposed a multiscale texture extraction method (Multiscale_Gabor_GLCM) that took advantages of multiscale Gabor transformation and gray-level co-occurrence matrix (GLCM) analysis. Narrowband normalized difference vegetation indices (NDVIs) involving all possible two-band combinations and continuum removal of red-edge spectra (SpeCR) were also extracted for biomass estimation. Subsequently, non-parametric linear (i.e., partial least squares regression, PLSR) and nonlinear regression (i.e., least squares support vector machine, LSSVM) analyses were conducted using the extracted spectral features, multiscale textural features and combinations thereof. The visualization technique of LSSVM was utilized to select the multiscale textures that contributed most to the biomass estimation for the first time. Compared with the best-performing NDVI (1193, 1222 nm), the SpeCR yielded higher coefficient of determination ($R^2$), lower root mean square error (RMSE), and lower mean absolute error (MAE) for winter wheat biomass estimation and significantly alleviated the saturation problem after biomass exceeded 800 g/m$^2$. The predictive performance of the PLSR and LSSVM regression models based on SpeCR decreased with increasing bandwidths, especially at bandwidths larger than 11 nm. Both the PLSR and LSSVM regression models based on the multiscale textures produced higher accuracies than those based on the single-scale GLCM-based textures. According to the evaluation of variable importance, the texture metrics "Mean" from different scales were determined as the most influential to winter wheat biomass. Using just 10 multiscale textures largely improved predictive performance over using all textures and achieved an accuracy comparable with using SpeCR. The LSSVM regression model based on the combination of the selected multiscale textures, and SpeCR with a bandwidth of 9 nm produced the highest estimation accuracy with $R^2_{val}$ = 0.87, RMSE$_{val}$ = 119.76 g/m$^2$, and MAE$_{val}$ = 91.61 g/m$^2$. However, the combination did not significantly improve the estimation accuracy, compared to the use of SpeCR or multiscale textures only. The accuracy of the biomass predicted by the LSSVM regression models was higher than the results of the PLSR models, which demonstrated LSSVM was a potential candidate to characterize winter wheat biomass during multiple growth stages. The study suggests that multiscale textures derived from high-definition UAV-based digital images are competitive with hyperspectral features in predicting winter wheat biomass.

**Keywords:** UAV digital images; winter wheat biomass; multiscale textures; red-edge spectra; least squares support vector machine; variable importance

## 1. Introduction

As an important grain crop, sustainable wheat production plays a crucial role in ensuring food security in the context of rapidly increasing population worldwide. Above-ground biomass (AGB) usually serves as an important factor in crop growth monitoring and grain yield prediction due to its strong relevance with crop growth status and soil fertility. Moreover, the information on crop biomass is required in calculating critical nitrogen concentration and nitrogen nutrition index, which can well indicate crop nitrogen status and thus facilitate the adjustment of in-season nitrogen management [1,2]. Conventional measurements of crop biomass mainly involve massive manpower investment and destructive sampling in the field [3]. These ground-based methods are time-consuming and not suitable for large-scale crop monitoring, since most of agronomical parameters exhibit high spatial variability within field [4]. Therefore, it is imperative to develop advanced and effective technologies to timely and economically obtain crop biomass information during growing seasons over large areas.

Over the past few decades, remote sensing technology has been gradually recognized as an effective and non-invasive tool for estimating crop biomass and characterizing its spatial and temporal variability. Compared with synthetic aperture radar (SAR) and light detection and ranging (LiDAR), optical remote sensing is still an attractive choice to obtain crop biomass information due to its relatively low cost and the availability of new optical sensors with fine spectral, temporal, and spatial resolutions. The vegetation indices (VIs) derived from multispectral broadband remotely sensed data have long been used as indirect measurements of crop biomass, such as normalized difference vegetation index (NDVI), enhanced vegetation index (EVI), and normalized difference water index (NDWI) [5–7]. However, these vegetation indices have limited value and often lose sensitivity for dense crop canopies when biomass exceeds a certain amount. This VI-AGB insensitivity is known as the saturation problem. Recent studies have asserted that hyperspectral remotely sensed data have great potential to overcome the saturation problem in crop biomass estimation, because they offer approximately continuous spectra and can describe vegetation canopy reflectance in greater detail as compared to multispectral data [3,8–10]. The red-edge region, the transition of red-near infrared, has been shown to have high information content for crop biomass. In the study of Hansen and Schjoerring [3], the newly constructed NDVI (718, 720 nm) yielded high accuracy for the estimation of winter wheat biomass. Fu et al. [9] extracted sensitive features from red-edge spectra using band-depth analysis and combined them with partial least squares regression to estimate winter wheat biomass, achieving higher accuracy than did the optimal NDVI (1097, 980 nm). Marshall and Thenkabail [10] also highlighted the importance of red-edge spectra when comparing the predictive performance of multispectral broadband and hyperspectral narrowband spaceborne remote sensed data for crop biomass estimation. Apart from the wavebands in red edge, several wavebands located in the near-infrared region have been reported to be sensitive to crop biomass. Yao et al. [11] indicated that the continuous wavelet feature located at 1197 nm yielded more accurate estimates of wheat biomass than the existing narrowband hyperspectral VIs. Fu et al. [9] evaluated the predictive performance of all possible two-band combinations in the form of NDVI for winter wheat biomass estimation and found that the optimal NDVI was calculated by the wavebands 1097 and 980 nm. Marshall and Thenkabail [10] indicated that three near-infrared wavebands of Hyperion EO-1 hyperspectral imagery including 914, 1130, and 1320 nm were crucial for crop biomass estimation. In spite of the appealing characteristics of hyperspectral data, many studies associated with hyperspectral remote sensing of crop biomass utilized hand-held devices or ground platforms to acquire hyperspectral data. They are not applicable to crop biomass

estimation at large scale. With the accomplishment of upcoming hyperspectral spaceborne missions, the situation will change, and vast hyperspectral data will be available for crop monitoring. However, the weather conditions and specific characteristics of satellite sensors largely limit the availability of high quality hyperspectral images, which satisfy the requirements of precision crop monitoring.

Currently, the rapid development of unmanned aerial vehicles (UAVs) and compact optical sensors opens a new perspective for quantifying crop biomass. The UAV remote sensing systems have shown great potential to fill the gap between ground-based devices and spaceborne and manned airborne remote sensing systems. Compared with ground-based platforms, UAV platforms were no longer confined to a small survey area and can largely improve the efficiency of data collection. The relatively low-cost UAV systems were more accessible for researchers and farmers than spaceborne and manned airborne systems. Moreover, the data acquired by UAV systems usually have higher temporal and spatial resolutions as compared with the existing spaceborne remote sensing systems. A variety of imaging data including red green blue (RGB) digital, multispectral, and hyperspectral images acquired from UAV platforms have been exploited in crop biomass estimation. Bendig et al. [12] indicated that barley biomass was strongly correlated with canopy height, which was extracted from point clouds generated by UAV-based RGB imaging. However, these photogrammetry-derived plant heights were not reliable for dense crop canopies [13]. Moreover, the change in canopy height was not obvious during late growth stages. Therefore, it is not feasible to estimate crop biomass only based on plant height during the entire growth period. Lu et al. [14] reported that the combination of VI and canopy height derived from UAV-based RGB images produced higher estimation accuracy of wheat biomass than the use of VI or canopy height alone. However, the VIs derived from UAV-based RGB images also suffered from the saturation problem. To obtain accurate estimates of potato aboveground biomass, Li et al. [15] established a random forest regression mode, which combined canopy height derived from UAV-based RGB images and multiple hyperspectral indices calculated from UAV-based hyperspectral images. Of these UAV remote sensing systems, the UAV systems equipped with an RGB digital camera have gained great popularity in precision agriculture, owing to its low cost and easy integration with other agricultural management tools. Due to the limitation of UAV-based RGB VIs and canopy height in crop biomass estimation, researchers have begun to mine spatial features from ultrahigh-spatial-resolution RGB images (<10 cm). Texture, a typical spatial feature, is expected to be more effective in imagery with finer spatial resolution, since more discriminative structural details can be shown [16]. Zheng et al. [17] proposed the normalized difference texture index (NDTI), which was calculated by the gray level co-occurrence matrix (GLCM)-based textures. Their experimental results demonstrated the superiority of the NDTI to the evaluated multispectral VIs for the estimation of rice biomass. In the study of Yue et al. [18], the GLCM-based textural features were extracted from UAV-based RGB images with five different spatial resolutions and exhibited better predictive performance in the wheat biomass estimation as compared with traditional hyperspectral VIs and RGB-based VIs. Commonly, crop canopy size and plant density vary with growth stages and field management practices. Therefore, it is difficult to characterize the spatial changes of crop canopy using textural features with a single scale. Better crop estimates may be possible if multiscale textures can be well extracted. Motivated by this idea, a multiscale image texture extraction method is proposed in the study. The proposed texture extraction method takes good advantage of Gabor filters and GLCM analyses. Gabor filters can provide multiscale and multidirectional spatial features that are tolerant to local illumination changes, while GLCM-based textures have the capability of characterizing the local heterogeneity in the tone values of pixels as well as their spatial distribution. To the best of our knowledge, the predictive performance of multiscale textural features has rarely been evaluated in crop biomass estimation.

It is critical to employ appropriate regression methods to correlate the extracted features with crop biomass. Recently, non-parametric regression methods have exhibited superiority over parametric regression methods in crop traits retrieval, especially machine learning based regression techniques. Studies are now saturated with multiple examples of machine learning regression algorithms to estimate a variety of crop traits from remotely sensed data [19–21]. However, it is less informative to treat the established machine learning regression model solely as a black box. Remotely sensed data analyses should always push toward better understanding of the underlying mechanisms and identification of the important spatial and spectral features that contribute most to crop trait of interest. An extensive interest has been shown in applying kernel-based machine learning regession algorithms for crop trait estimation, because they have high generalization performance and good ability to characterize non-linear relationships in a global manner. Owing to the study of Ustun et al. [22], the kernel-based machine learning regression (e.g., support vector machine regression and kernel partial least squares regression) can be interpreted like partial least squares regression (PLSR) rather than a black box technique. The proposed visualization method makes it possible to interpret the driving force behind the established kernel-based regression model. Least squares support vector machine (LSSVM), as a variant of SVM, also applies structural risk minimization and is superior to artifical neural networks (ANNs) in terms of model generalization when limited training samples are available [23]. At present, only limited studies have used LSSVM to retrieve vegetation biophyiscal parameters [24,25]. Moreover, the interpretation of LSSVM model has rarely been conducted. In this study, PLSR and LSSVM were selected as representatives of non-parametric linear and nonlinear regression techniques and used to model wheat biomass. Hyperspectral features describe the average tonal variations in multiple wavebands, whereas textural features contain information concerning the spatial distribution for tonal variations winthin a waveband [26].

The goal of this study was to investigate if the proposed multiscale textures extracted from UAV digital images could compete with hyperspectral information for winter wheat biomass estimation. The specific objectives of this study were to (1) compare the predictive performance of optimal NDVI and continuum removal of red-edge spectra with different bandwidths; (2) evaluate the effectiveness of the proposed multiscale texture extraction method (hereafter referred to as Multiscale_Gabor_GLCM) and determine the optimal textural features for winter wheat biomass estimation; and (3) demonstrate if the combination of multiscale textures and spectral features (i.e., optimal NDVI and continuum removal of red-edge spectra) could further improve estimation accuracy.

## 2. Materials and Methods

### 2.1. Experimental Design

Winter wheat field experiments were conducted during three growing seasons (2014–2015, 2017–2018, and 2018–2019) at the Xiaotangshan Experiment Site (40.175°~40.188°N, 116.436°~116.451°E, geographic WGS84), Changping district, Beijing, China. The soil is classified as fine loam with an organic content of 15.3~19.1 mg/kg, a nitrate-nitrogen (NO3-N) content of 8.21~41.71 mg/kg, an ammonium-nitrogen (NH3-N) content of 0.45~8.2 mg/kg, an available potassium content of 60.58~109.61 mg/kg, and an available phosphorus content of 3.14~21.18 mg/kg at top 30 cm soil layer. This experimental station is characterized by a warm temperate and semi-humid continental monsoon climate. The minimum and maximum temperature are $-10$ °C and 40 °C, respectively. Most of precipitation occurs in summer and the annual amount is about 620 mm.

The trails were undertaken in experimental plots and considered three experimental variables included different cultivars, irrigation rates, and nitrogen (N) application levels (Table 1). Experiment 1 (Exp. 1) followed an orthogonal experimental design and involved three replications of two varieties (Zhongmai 175 and Jing 9843), four N application rates, and three irrigation levels. In this experiment, forty-eight sampling plots were designed with each covering an area of 48 m$^2$ (6 m × 8 m). Exp. 2 and Exp. 3 were conducted in

two consecutive growing seasons (2017–2018 and 2018–2019), and both of them followed a randomized complete block design. There were 64 and 32 sampling plots for the two experiments, respectively. The covering area of each plot was about 135 m$^2$ (9 × 15 m). The two experiments conducted four repetitions and considered two winter wheat varieties (Lunxuan 167 and Jingdong 18) and four N application rates. The detailed experiment designs can be referred to the studies of Xu et al. [27] and Fu et al. [20]. For all experiments, 50% nitrogen fertilizer was applied prior to seeding, and the left was applied at the jointing stage. Other field management practices followed the local farmlands for winter wheat production.

**Table 1.** Summary of the treatments used in the three winter wheat field experiments.

| Growing Season | Cultivar | Sowing Date | Treatments | Sampling Period |
|---|---|---|---|---|
| Exp. 1: 2014–2015 | Zhongmai175, Jing 9843 | 7 October | N rate (kg N/ha): 0, 90, 180, 270; Irrigation rate (mm): 0, 192, 384 | 14 April, jointing (Z.S. 31); 26 April, booting (Z.S. 47); 12 May, anthesis (Z.S. 65); 26 May, filling (Z.S. 75) |
| Exp. 2: 2017–2018 | Lunxuan 167, Jingdong 18 | 30 September | N rate (kg N/ ha): 0, 90, 180, 270 | 3 May, booting (Z.S. 47); 16 May, anthesis (Z.S. 65) |
| Exp. 3: 2018–2019 | Lunxuan 167, Jingdong 18 | 5 October | N rate (kg N/ ha): 0, 90, 180, 270 | 16 April, jointing (Z.S. 31); 30 April, booting (Z.S. 47) |

Note: Z.S. is the abbreviation for Zadoks growth stage.

*2.2. Data Collection*

2.2.1. UAV Images Acquisition and Pre-Processing

For Exp. 1 and Exp. 2, an eight-rotor UAV equipped with a high-definition digital camera was used to acquire RGB digital images of winter wheat canopy. The camera (Sony DSC-QX100, Sony, Tokyo, Japan; 20.2 megapixel; 64° field of view) was attached to a gimbal, which could compensate the UAV movement during the flight and guaranteed nearly nadir image acquisition. For Exp. 3, a DJI phantom 3 with embedded RGB digital camera (1/2.3-inch CMOS; 12 megapixel) was used to capture images of winter wheat canopy. The UAV systems flew autonomously over the study area according to the predefined flight route programmed by the DJI ground station. The longitudinal and lateral overlaps were set to 80% and 75%, respectively. The flight altitude was about 50 m, corresponding to a ground sampling distance of about 1 cm. The integration time of cameras was adjusted automatically according to the light conditions of each flight to avoid over-saturation and ensure the maximum dynamics. The camera focal length was 10 mm and the aperture was set to f/11 and f/3.2 for Exp. 1 and Exp. 2, respectively. The camera focal length was 9 mm, and the aperture was set to f/5.6 for Exp. 3. All flights were accomplished under cloud-free daylight conditions between 10:00 a.m. and 2:00 p.m. (Beijing local time).

Prior to extracting informative features from UAV-based RGB imagery, image-preprocessing must be conducted, and the main procedures included aligning images, generating dense point cloud, building mesh, constructing texture, and mosaicking image. The generation of ortho-rectified and geo-referred mosaic image depends on the structure from motion algorithm and mosaic blending theory [28,29]. We co-registered the RGB orthomosaics acquired at different growth stages using the image-to-image registration workflow of ENVI (Exelis Visual Information Solution, Boulder, CO, USA) with a geometric accuracy of <0.5 pixel. To maintain the radiometric consistency of multitemporal remotely sensed images, relative radiometric correction was conducted following the algorithm in the literature [30]. The orthomosaic images for the three experiments were all resampled to a ground sampling distance of 1 cm through nearest-neighbor interpolation. To further avoid the influence of illumination variation, the normalization was implemented through dividing each pixel value by the corresponding sum of the pixel value of all bands [31]. For each sampling plot, a region of interest (ROI) was selected from the center of plot

area to exclude border effect. Consequently, a total of 384 ROIs were cropped, including 192 ROIs (448 × 553 pixels) for Exp. 1, 128 ROIs (621 × 930 pixels) for Exp. 2, and 64 ROIs (621 × 930 pixels) for the Exp. 3.

### 2.2.2. Hyperspectral Data Acquisition and Biomass Measurements

After UAV flight missions, the field hyperspectral reflectance measurements of wheat canopy were acquired by a field spectroradiometer (Analytical Spectral Devices, Boulder, CO. USA) with a spectral range of 350~2500 nm under cloud-free conditions around solar noon. The spectral resolution is 1.4 nm between 350 and 1050 nm, while the spectral resolution is 2 nm between 1050 and 2500 nm. Spectral data were usually interpolated to a spectral resolution of 1 nm for further analysis by the corresponding software. The sensor was placed vertically approximately 1.3 m above wheat canopy, with a field of view of 25°. Prior to each measurement, the radiance of a white standard spectral reference panel was recorded to convert the radiance measurements of each plot to reflectance. To ensure the reliability of spectral measurments, ten replicate spectra were taken for each plot, and the average spectrum of them was used to represent the spectrum of the plot. The spectra in the spectral region of 400~1350 nm were used for the subsequent analyses, due to their high signal to noise ratio. To further suppress noise in the spectral measurements, a Savitzky–Golay filter with a frame size of 19 was used.

After the collection of hyperspectral data, ground-based plant sampling was undertaken, and an area of 0.25 $m^2$ wheat plants was clipped at ground level for each plot. The cut plants were sealed in a plastic bag and sent to laboratory immediately. These samples were firstly green-killed in an oven at 105 °C for about 30 minutes and then dried at 80 °C until a constant weight. The dry aboveground biomass was expressed on the basis of ground area according to the number of plant stems per unit area.

### 2.3. Multiscale Textures Extraction by Fusing Gabor Filter and GLCM Analyses

To capture the changes in winter wheat canopy during growing season well, multiscale textural features derived from high-definition UAV-based RGB images were required and expected to be more suitable for biomass estimation. To this end, we proposed a new texture extraction method, which took advantages of two-dimensional (2D) Gabor filter and GLCM analyses. For illustrative purposes, Figure 1 shows the schematic of the proposed multiscale texture extraction method (Multiscale_Gabor_GLCM). GLCM is computed for various angular relationships and distances between nearest pixel pairs in the image [26]. The textures derived from GLCM can well represent the spatial distribution of gray tones rather than the solely tones. The human visual system is regarded to have a special function to process information in multiresolution levels, which can be mimicked by changing the scale parameter in 2D Gabor filters [32]. Gabor filters are actually a series of local spatial bandpass filters, which are characterized by good spatial localization, orientation selectivity, and frequency selectivity [32]. In this study, four orientations ([0°, 45°, 90°, 135°]) and five scales ($v \in [0, 1, 2, 3, 4]$) were considered to generate a bank of Gabor filters, which approximately uniformly covered the spatial-frequency domain. The settings of other parameters were the same with the literature [20]. According to our previous study [20], for UAV-based RGB images of winter wheat canopy, there was no significant heterogeneity in both Gabor- and GLCM-based textures of different orientations, and the textures of 45° were recommended. Thus, we only considered the orientation of 45° in the subsequent Gabor representation and GLCM analyses. For a ROI of winter wheat, each image band was convoluted with the Gabor filter bank, and a total of 15 Gabor magnitude images were generated. Then GLCM-based textures were extracted from each Gabor magnitude image to form multiscale image textural features. The used GLCM-based textures mainly included mean (Mean), variance (Var), homogeneity (Hom), contrast (Con), dissimilarity (Dis), entropy (Ent), second moment (Sec), and correlation (Cor) [26]. Consequently, a total of 120 textural features were extracted for each ROI image.

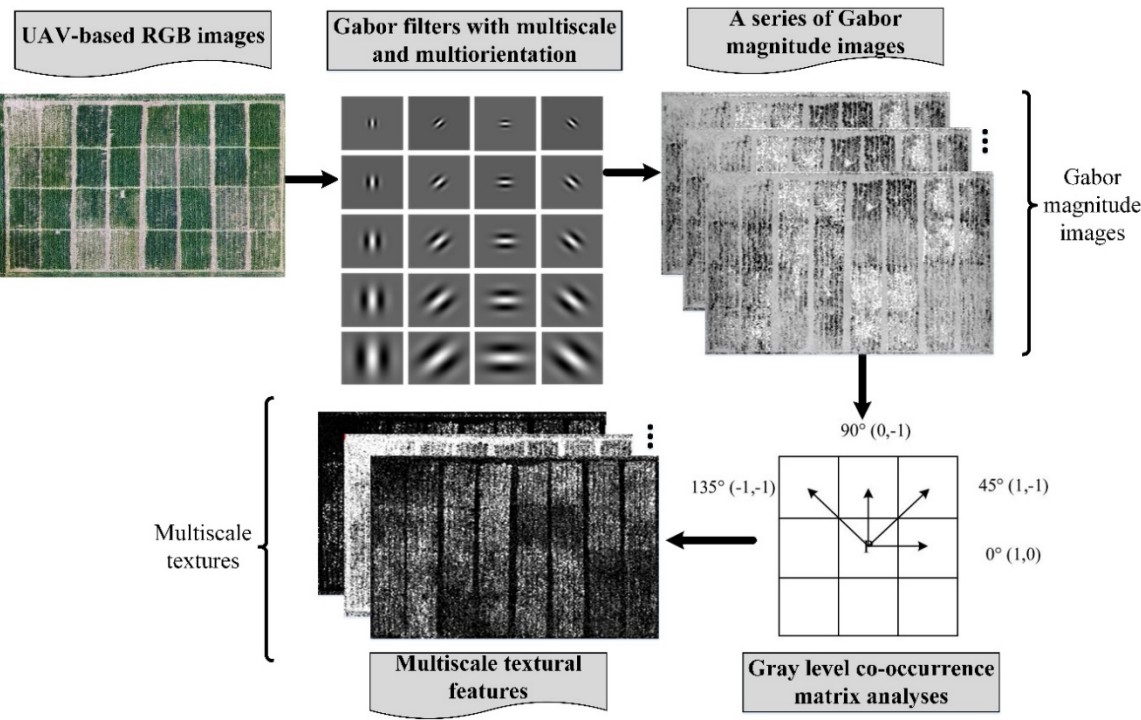

**Figure 1.** System block diagram of the proposed multiscale texture extraction method (Multiscale_Gabor_GLCM).

The suffixes indicating specific feature scale and image band followed the names of textures for convenience and illustrative purpose. For instance, "Hom-G-S2"represents the texture homogeneity at scale 2 derived from green band, "Sec-R-S1"represents the texture second moment at scale 1 derived from red band, and "Cor-B-S5"represents the texture correlation at scale 5 derived from blue band.

### 2.4. Biomass Related Hyperspectral Feature Extraction

To extract hyperspectral features sensitive to wheat biomass, two methods were conducted including optimal narrowband vegetation index and continuum removal of red-edge spectra.

#### 2.4.1. Narrowband Vegetation Index

Two-band normalized difference vegetation index (NDVI) was used to select biomass specific wavebands. All possible two-pair combinations of 951 wavelengths were used to calculate the narrowband NDVI-like indices. Then, the coefficient of determination ($R^2$) was obtained in the univariate linear regression of these NDVI-like indices and winter wheat biomass based on the calibration dataset. According to the two-dimension (2D) correlation scalogram, the wavelength combinations resulting in the highest $R^2$ values formed the optimal NDVI-like indices and were identified as sensitive spectral features to winter wheat biomass.

#### 2.4.2. Continuum Removal of Red-Edge Spectra with Different Bandwidths

Red-edge spectral region has proved to have abundant information for crop biomass. To effectively extract spectral features from red edge, continuum removal was conducted on the spectra between 550 and 750 nm. Comparing with original hyperspectral data, continuum removal enables direct comparison of individual absorption features from a common baseline. Considering the strong intercorrelation among adjacent hyperspectral wavebands, the spectra between 550 and 750 nm under different bandwidths were simulated with full width at half maximum (FWHM) ranging from 1 to 21 nm and a step of 2 nm.

The spectral response function was simulated by Gaussian function (*f*) and calculated with the following Equations (1) and (2).

$$f(b, \sigma) = \exp\left(-\frac{(-b - b_c)^2}{2\sigma^2}\right) \tag{1}$$

$$\sigma = \frac{FWHM}{2\sqrt{2ln2}} \tag{2}$$

where *b* represents the wavelength in the spectral response range for a certain band width; $b_c$ is the central wavelength of interest; $\sigma$ is standard variance; and *ln* is natural logarithm operator.

Apart from the reduction of the hyperspectral dimension, simulating spectra with different bandwidths was conducive to investigating the influence of bandwidth on the predictive performance of red-edge spectral features for winter wheat biomass estimation.

### 2.5. Non-Parametric Modeling Methods: PLSR and LSSVM

In this study, two different regression techniques were analyzed, including a non-parametric linear regression (i.e., PLSR) and a non-parametric nonlinear regression (i.e., LSSVM), so that improvements in estimation accuracy of features–algorithm pairs could be evaluated. Moreover, the visualization of LSSVM facilitated the identification of influential textural features for winter wheat biomass and improved the understanding of the established predictive models.

#### 2.5.1. Partial Least Squares Regression

As a computationally efficient modeling algorithm, PLSR has been widely used to establish predictive models of crop traits based on a variety of features, due to its ability to cope with multi-collinearity and high-dimensional data. Rather than vegetation indices only using several wavebands, PLSR can incorporate full-spectrum data into modeling analysis through decomposing high-dimensional data into several orthogonal latent factors as well as maximizing the covariance between dependent and independent variables [33]. The leave-one-out cross-validation (LOOCV) was used to select the optimal latent factors, which is crucial to model performance. To make compromises between model predictive ability and simplicity, the criterion of adding an extra factor to the model was that the root mean square error of LOOCV was decreased by no less than 2% [34,35].

#### 2.5.2. Least Squares Support Vector Machine

As a variant of SVM, LSSVM inherits many strengths of standard SVM, including application of structural risk minimization, being capable of processing data with high-dimension and non-normal distribution, and overcoming the problem of small sample [23]. However, the solution of LSSVM is accomplished through solving a set of equality constraints instead of inequality ones in standard SVM [23]. It avoid solving a quadratic programming and largely improves the effeciency of model establishement. LSSVM has been widely adopted in characterizing the nonlinear relationships between crop traits and various features. However, the driving force behind the established LSSVM regression models has been rarely interpreted and visualized. In this study, we took advantage of the visualization method for support vector regression (SVR) and investigated the influential textures for winter wheat biomass estimation. In the study of Ustun et al. [22], the interpretation of a SVR model is realized by computing the inner product of the independent variable matrix and the vector of Lagrange multipliers ($\alpha$-vector). The length of the resulting new vector is the number of independent variables. This new vector can be used to indicate which independent variables are highly correlated to the dependent variables of interest, like the partial least squares (PLS) loadings or regression coefficients. After identifying the most relevant independent variables through the initial LSSVM regression model, the final LSSVM regression was rebuilt with relevant independent variables. During the

LSSVM regression modeling, the regularization parameter and kernel function parameters such as the standard variance of radial basis function (RBF) kernel were determined by LOOCV in the study.

### 2.6. Modeling Framework and Validation

To construct and validate winter wheat biomass predictive models, the three-year experimental data were split into calibration and validation datasets. The calibration dataset was used to adjust the parameters of models, while the validation dataset was used to confirm the true predictive power of the established models and had no effect on model construction. The validation dataset consisted of the data from the first replicate collected in 2015 and the fourth replicate collected in 2018 and 2019. The left data made up the calibration dataset. Consequently, there were 272 and 112 samples in calibration and validation datasets, respectively. Figure 2 illustrates the modeling framework of this study. To validate the effectiveness of the proposed texture extraction method Multi-scale_Gabor_GLCM, the GLCM-based textures were directly extracted from UAV-based RGB images and used to construct winter wheat biomass predictive model. To illustrate the necessity of the extraction of multiscale textures, the predictive performance of the textures from each scale was evaluated. The spectral-based predictive models included the linear regression model based on the optimal NDVI-like and the models based on the continuum removal of red-edge spectra with different bandwidths. The PLSR and LSSVM analyses based on the combination of multiscale textures and biomass related spectral features were conducted to ascertain whether the combination could improve estimation accuracy. Before modeling analyses, the combined feature matrices were standardized to reduce the influence of different ranges of feature values. The coefficient of determination ($R^2$), root mean squared error (RMSE), and mean absolute error (MAE) between observations and estimations were used to assess the accuracy of biomass predicted by different models.

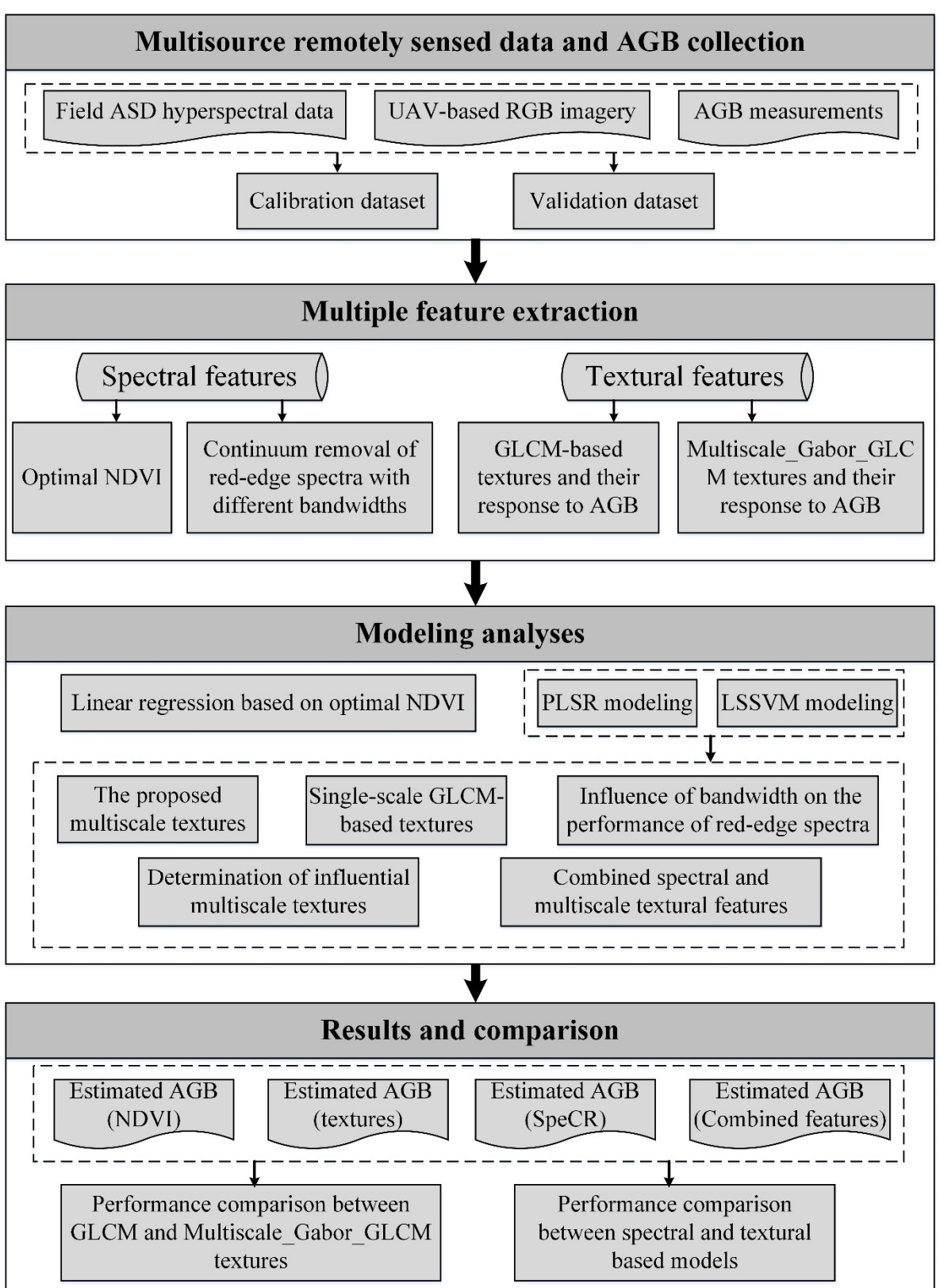

**Figure 2.** Modeling framework and comparison analyses of the study.

## 3. Results

### 3.1. Descriptive Statistics

The differences in winter wheat aboveground biomass were created by multiple treatments including different varieties, different levels of N fertilizer applications, and different irrigation rates. The AGB values under different treatments were compared through the least significant different test (LSD) at the 95% level of significance. The results showed that there existed significant difference in the winter wheat AGB collected from the three experiments ($p < 0.05$). The descriptive statistics of winter wheat AGB exhibited a high degree of variation with a coefficient of variation (CV) of 57.88% for the calibration dataset and a CV of 60.88% for the validation dataset (Table 2). The range of AGB in the calibration dataset is larger than that for the validation dataset. The kurtosis values indicated that the AGB in both calibration and validation datasets approximately followed normal distribution. To help to understand the RMSE and MAE for calibration and validation, the minimum (Min.), mean, maximum (Max.), and standard deviation (Std.) of the observed AGB are also shown in Table 2.

**Table 2.** Descriptive statistics for winter wheat AGB ($g/m^2$) of calibration and validation datasets.

| Wheat AGB | Min. | Mean | Max. | Std. | CV (%) | Kurtosis |
|---|---|---|---|---|---|---|
| Calibration | 50.06 | 575.85 | 1759.95 | 333.30 | 57.88 | 3.38 |
| Validation | 75.96 | 526.17 | 1501.63 | 320.38 | 60.88 | 3.25 |

### 3.2. Using Spectral Features to Estimate Winter Wheat AGB

#### 3.2.1. Univariate Linear Regression Based on Narrowband NDVI-Like

Narrowband NDVI-like indices were calculated using all possible two-band combinations. The univariate linear regression analyses of winter wheat biomass against these NDVI-like indices were performed to obtain the $R^2$ values. As shown in Figure 3a, the formed two-dimension (2D) correlation scalogram exhibits a characteristic pattern with multiple "hot spots" with relatively high $R^2$ values. The spots were selected by choosing the wavelength combination that had an $R^2$ larger than 0.55 ($p < 0.01$). It could be observed that most of the wavebands forming these "hot spots" were located in the spectral range of 1168~1276 nm, and several band pairs were located in the spectral range of 675~683 nm. The optimal band combination with the largest $R^2$ was composed of 1193 and 1222 nm. The univariate linear regression model based on the optimal NDVI-like (1193, 1222 nm) indices was constructed and assessed using the validation dataset. The predictive performance of this model is shown in Figure 3b. The RMSE of this model in the validation dataset reached up to 197.30 $g/m^2$. The majority error resulted from the underestimation for the biomass samples exceeding 800 $g/m^2$. It indicated that the optimal NDVI-like (1193, 1222 nm) suffered from saturation problem and consequently resulted in reduced winter wheat AGB estimation accuracy.

#### 3.2.2. Non-Parametric Modeling Based on the Continuum Removal of Red-Edge Spectra

To investigate the influence of bandwidth on the predictive performance of continuum removal of red-edge spectra (SpeCR), both PLSR and LSSVM models were established based on the SpeCR under different bandwidths. In the comparison of Figure 4a,b, it could be observed that the predictive performance of the LSSVM regression models was generally better than that of the PLSR regression models. The RMSE values for the validation dataset generally increased with increasing bandwidth (Figure 4a,b). When the bandwidth exceeded 11 nm, the increasing RMSE values indicated that both LSSVM and PLSR models had an obvious decrease in predictive performance. Interestingly, both PLSR and LSSVM analyses on the SpeCR with a bandwidth of 1 nm did not produce the highest estimation accuracy. The PLSR model based on the SpeCR yielded the highest accuracy when the bandwidth was 5 ($R^2_{val} = 0.83$, $RMSE_{val} = 133.45$ $g/m^2$, and $MAE_{val} = 101.65$ $g/m^2$).

The LSSVM regression model based on the SpeCR achieved the highest accuracy when the bandwidth was 9 ($R^2_{val}$ = 0.87, $RMSE_{val}$ = 123.98 g/m$^2$ and $MAE_{val}$ = 92.08 g/m$^2$). Compared with the linear model based on the optimal NDVI-like, the optimal models based on the SpeCR significantly improved the estimation accuracy and decreased the RMSE value by 63.85~73.32 g/m$^2$. As observed in the scatter plots between observed and estimated AGB (Figure 4c,d), to some extent, the models based on the SpeCR alleviated the underestimation problem for the AGB exceeding 800 g/m$^2$, especially the LSSVM regression model.

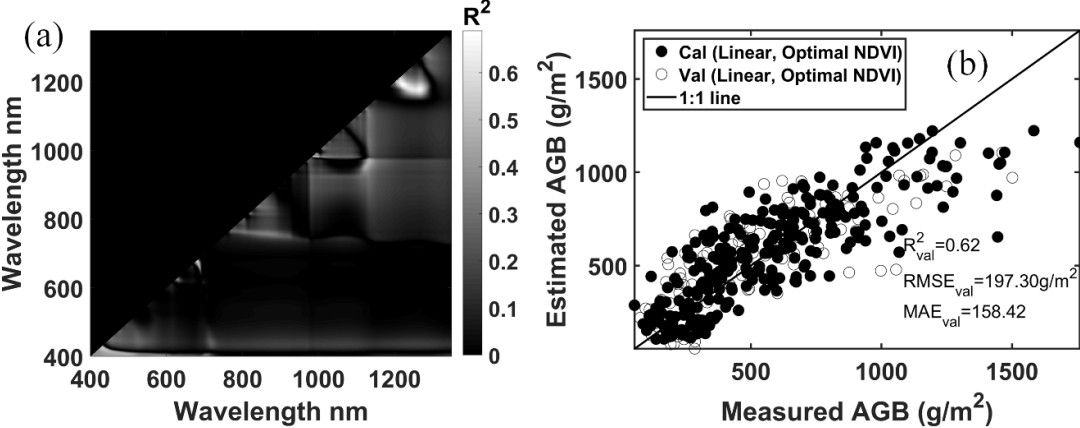

**Figure 3.** (**a**) Two-dimension scalogram illustrating the coefficient of determination ($R^2$) between biomass and narrowband NDVI-like indices based on all possible two-band combinations; (**b**) predictive performance of the optimal NDVI-like (1193, 1222 nm) for winter wheat biomass estimation.

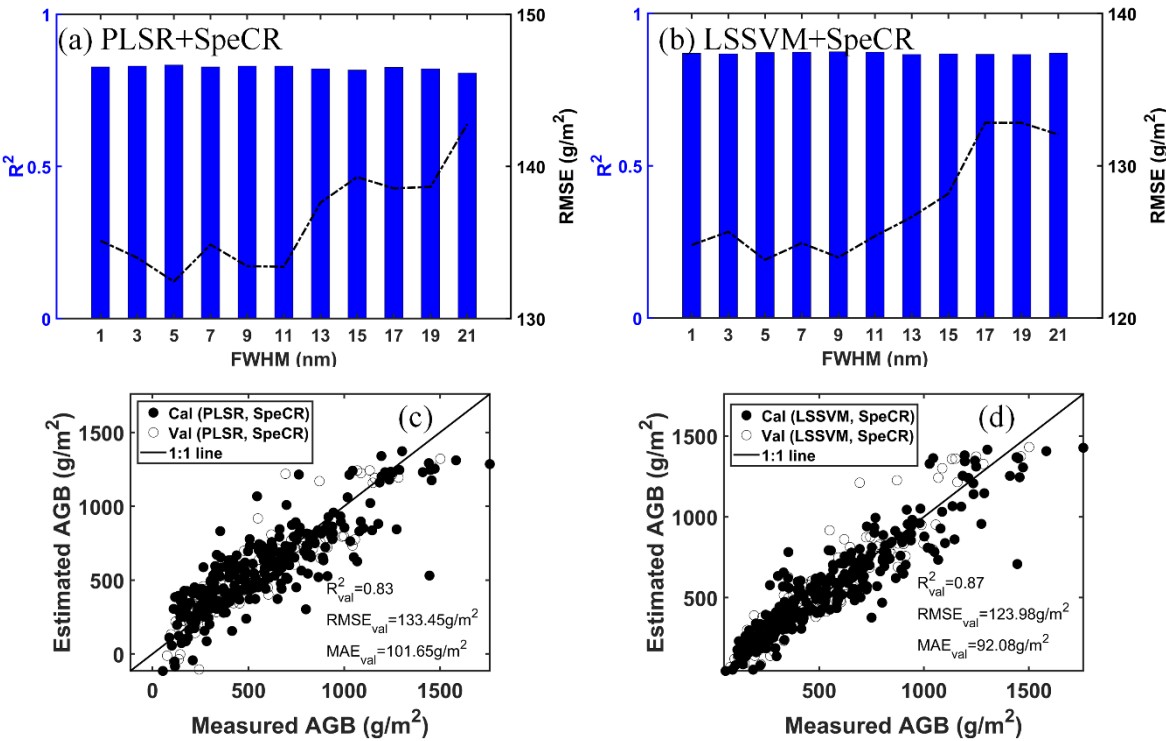

**Figure 4.** Predictive performance of the continuum removal of red-edge spectra (SpeCR) for winter wheat biomass estimation. Change of the predictive performance of SpeCR using PLSR (**a**) and LSSVM (**b**) under different bandwidths. Predictive performance of the SpeCR using PLSR at the optimal bandwidth (FWHM = 5) (**c**) and using LSSVM at the optimal bandwith (FWHM = 9) (**d**).

### 3.3. Using Multiscale Textures to Estimate Winter Wheat AGB

The Pearson correlation analyses were conducted between textures and winter wheat AGB during multiple growth stages (Figure 5). For the single-scale GLCM-based textures, the texures from blue band showed higher correlation with AGB than the textures from red and green bands. The texture Var-B produced the highest correlation coefficient (r = 0.41, $p < 0.01$). For the textures extracted by the proposed Multiscale_Gabor_GLCM method, the textures correlating well with AGB could be observed at each scale (Figure 5b). The texture Mea-G-S1 showed the strongest correlation with AGB (r = 0.69, $p < 0.01$). As can be observed from the scatter plots shown in Figure 5c,d, there exist linear relationships between these best-performing textures and AGB for each growth stage. However, the sensitivity of these textures to the variation in AGB decreased when considering multiple growth stages. In the comparison of Figure 5c,d, the best-performing texture Mea-G-S1 exhibited higher correlation with AGB than did the best-performing GLCM-based texture Var-B. For the AGB exceeding 800 g/m$^2$, the texture Mea-G-S1 still stayed sensitive to AGB, while the texture Var-B approximately approached a saturation level and disorderly response to AGB. Therefore, the models based on the multiscale textures might be more promising than those based on the single-scale GLCM-based textures.

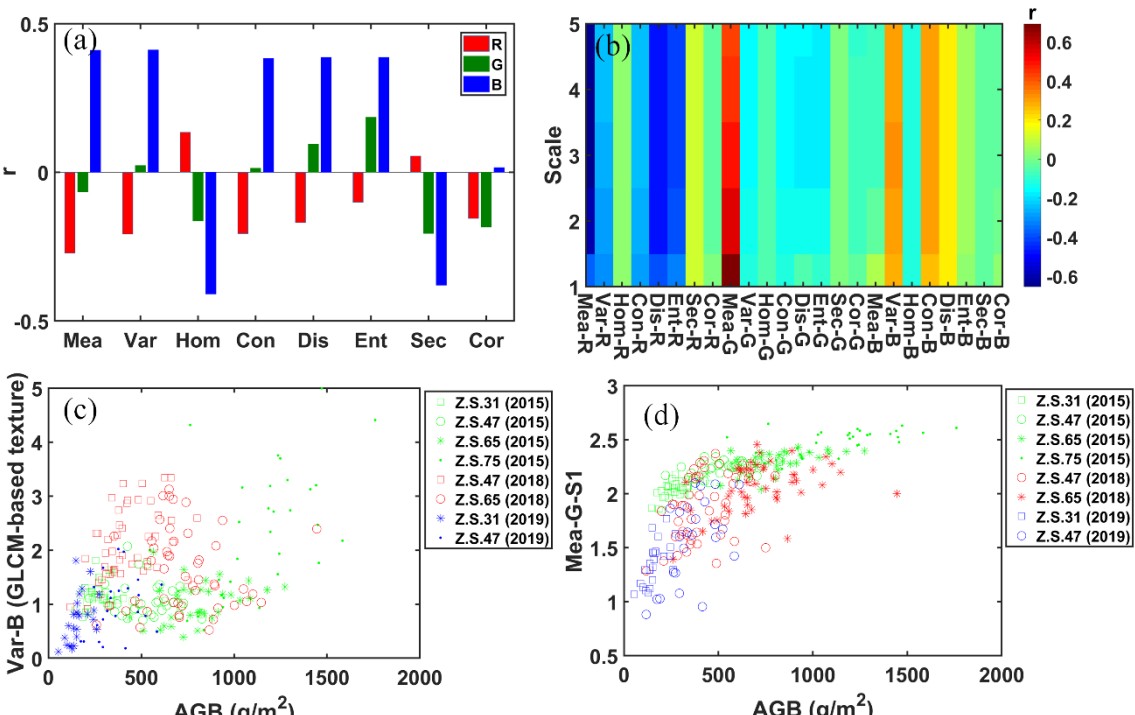

**Figure 5.** Responses of different textures to winter wheat AGB in the calibration dataset. (**a**) Pearson correlation analyses between GLCM-based textures and AGB; (**b**) Pearson correlation analyses between AGB and Multiscale_Gabor_GLCM textures; (**c**,**d**) Scatter plots between the best-performing textures and AGB for each growth stage.

PLSR analyses were conducted uisng the single-scale GLCM-based textures and multiscale textures, respectively. Figure 6 shows the results of these models for both calibration and validation datasets. For the two PLSR models, the optimal latent factors were 11 and 10, respectively. Compared with the PLSR model based on the single-scale GLCM-based textures, the PLSR model based on the multiscale textures performed better with a R$^2$ value of 0.78, a RMSE value of 154.54 g/m$^2$, and a MAE value of 122.86 g/m$^2$. This model increased the R$^2$ value by 0.09 and decreased the RMSE value by 24.58 g/m$^2$.

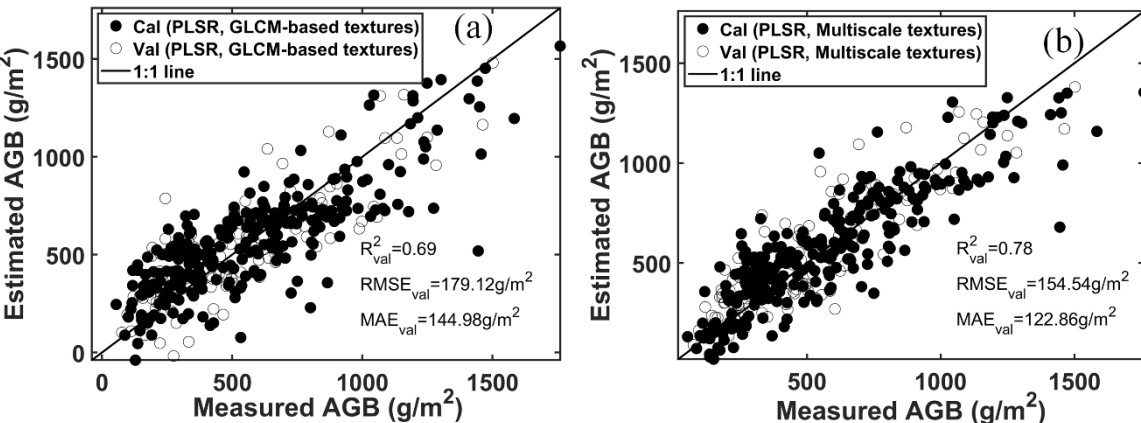

**Figure 6.** Predictive performance of the PLSR models using (**a**) the single-scale GLCM-based textures and (**b**) multiscale textures for winter wheat AGB estimation.

The influential textures were determined during the construction of LSSVM regression models. According to the importance value of each texture (Figure 7a for the GLCM-based textures; Figure 7b for the multiscale textures), the GLCM-based textures with importance values larger than 5 and the multiscale textures with importance values larger than 200 were selected for the rebuilding of new LSSVM regression models after several trials. A total of seven single-scale GLCM-based textures were selected including Mea-R, Var-R, Con-R, Mea-G, Mea-B, Var-B, and Con-B. The selected multiscale textures were Mea-R-S1, Mea-R-S3, Mea-R-S4, Mea-R-S5, Mea-G-S1, Mea-G-S2, Mea-G-S3, Mea-G-S4, Mea-G-S5, and Mea-B-S1 (10 textures). It can be observed that the multiscale texture metric "Mea" played an important role in AGB estimation. Compared with the initial LSSVM regression models based on all textures, the new LSSVM regression models not only improved the predictive performance, but also reduced the complexity of models (See Figure A1 in Appendix A). The LSSVM regression model based on the selected multiscale textures exhibited higher accuracy with $R^2_{val} = 0.85$, $RMSE_{val} = 129.45$ g/m$^2$, and $MAE_{val} = 97.11$ g/m$^2$, compared to the LSSVM regression model using the selected single-scale GLCM-based textures (Figure 7c,d). For both single-scale GLCM-based textures and multiscale textures, the LSSVM regression models performed better than the PLSR models (Figures 6 and 7). For instance, compared with the PLSR model based on the multiscale textures, the LSSVM regression model using the selected multiscale features increased the $R^2$ value by 0.08 and reduced RMSE value by 25.09 g/m$^2$. It indicated that LSSVM regression was superior to PLSR in characterizing the nonliner relationship between textures and winter wheat AGB. Compared with the spectral features based models, the LSSVM regression model based on the multiscale textures performed better than the linear model based on the optimal NDVI, but just got comparable estimation accuracy with the LSSVM regression model based on the SpeCR with a bandwidth of 9.

To further demonstrate the necessity of extracting multiscale textures for AGB estimation under multiple growth stages, the LSSVM regression analyses were conducted using the textures at each scale to examine their predictive performance. According to the importance values for the textures at each scale, it could be observed that the texture metric "Mea" still had a higher degree of importance than the other texture metrics. Among the textures from five scales, the textures at scale 1 yielded the best estimation accuracy with $R^2_{val} = 0.82$, $RMSE_{val} = 140.95$ g/m$^2$, and $MAE_{val} = 109.10$ g/m$^2$. The textures at scale 2 and scale 3 got the intermediate estimation accuracy. The textures at scale 5 got the worst estimation accuracy and achieved comparable accuracy with the single-scale GLCM-based textures. It could be obviously observed that the LSSVM regression model based on the multiscale textures produced better estimation accuracy than that based on the single-scale textures.

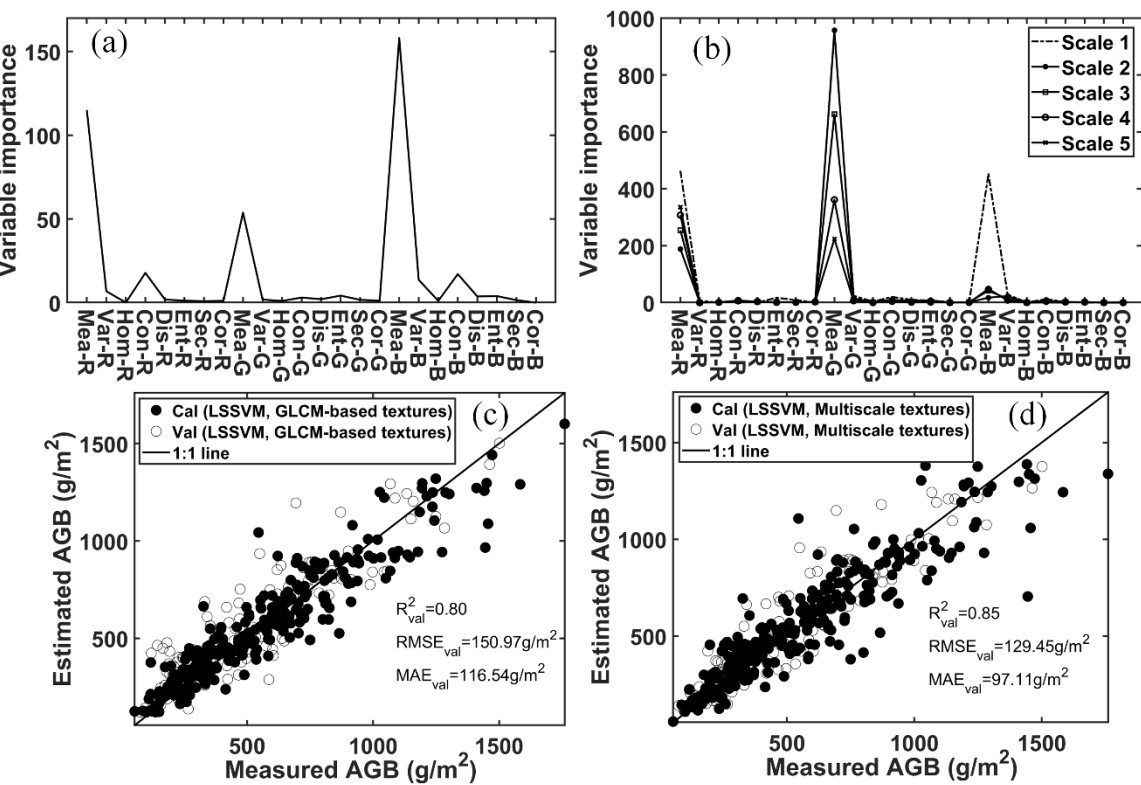

**Figure 7.** Predictive performance of the LSSVM regression models based on the selected GLCM-based textures (**a**,**c**) and selected multiscale textures (**b**,**d**) for winter wheat AGB estimation.

### 3.4. The Combined Use of Multiscale Textures and Spectral Features for Winter Wheat AGB Estimation

With combinations of selected multiscale textures, optimal NDVI and SpeCR, the LSSVM regression analyses did not significantly improve estimation accuracy and produced generally better (or similar) results than (or to) those using textures or SpeCR alone. Among these models, the LSSVM regression model based on the combination of selected multiscale textures and SpeCR (FWHM = 9) yielded the highest estimation accuracy of AGB with $R^2_{val}$ = 0.87, $RMSE_{val}$ = 119.76 g/m$^2$, and $MAE_{val}$ = 91.61 g/m$^2$ (Table 3). However, the combination of selected mulsticale textures and optimal NDVI just consisted of 11 features and yielded comparable estimation accuracy to the combination of multiscale textures and SpeCR (FWHM = 9) consisting of 29 features (i.e., 10 selected multiscale textures and 19 spectral features).

**Table 3.** Performance of the LSSVM regression models based on the combination of multiscale textures and biomass-sensitive spectral features for winter wheat biomass estimation.

|  | Calibration | | | Validation | | |
|---|---|---|---|---|---|---|
|  | $R^2$ | RMSE (g/m$^2$) | MAE (g/m$^2$) | $R^2$ | RMSE (g/m$^2$) | MAE (g/m$^2$) |
| Multiscale textures + optimal NDVI | 0.85 | 126.98 | 90.50 | 0.86 | 121.28 | 94.27 |
| Multiscale textures + SpeCR (FWHM = 9) | 0.87 | 121.89 | 85.85 | 0.87 | 119.76 | 91.61 |
| Multiscale textures + Optimal NDVI + SpeCR (FWHM = 9) | 0.87 | 120.55 | 84.59 | 0.87 | 121.69 | 91.77 |

### 4. Discussion

A three-year field experiment with different cultivars, N fertilizer application, and irrigation rates led to a large number of samples with high variations in winter wheat

AGB. An economical UAV remote sensing system successfully acquired the RGB digital imagery of winter wheat canopy during growing seasons. The spatial resolution of the RGB imagery reached up to 1 cm. It allowed the extraction of fine textural features and the assessment of their predictive performance for winter wheat AGB estimation. The ground-based hyperspectral data and biomass measurements were concurrently collected. A comparison of the predictive performance of hyperspectral features, textural features, and combinations thereof for winter wheat AGB estimation was also possible.

*4.1. Performance Comparison of Simple and Complex Spectral Models for Wheat Winter AGB Estimation*

Using all the possible two-band combinations for narrowband NDVI-like calculation facilitated the selection of sensitive hyperspectral features for winter wheat AGB. In the linear regression analyses of narrowband NDVI-like indices against winter wheat AGB, it was found that the wavebands forming the "hot spots" in the 2D correlation matrix plot were mainly located between 1168 and 1276 nm (Figure 2). This spectral area is close to 1200 nm, which is affected by the absorption of water, cellulose, starch, and lignin [36]. Yao et al. [11] also indicated that the continuous wavelet features extracted from this spectral region exhibited strong correlation with wheat AGB. Gnyp et al. [37] developed a new hyperspectral vegetation index named GnyLi, which was calculated from wavebands 900, 955, 1050, and 1220 nm. They indicated that the GnyLi vegetation index correlated well with winter AGB during several growth stages (Z.S.30~45) and highlighted the ability of local peak spectra in near-infrared and shortwave infrared regions in alleviating the saturation problem for vegetation AGB estimation. In our study, the best-performing NDVI-like index was composed of wavebands 1193 and 1222 nm. However, in the study of Fu et al. [9], the best narrowband NDVI-like index for winter wheat AGB estimation was calculated from wavebands located at 980 and 1097 nm. The inconsistency in selected wavebands mainly ascribed to the use of different calibration datasets. It is a common problem in band selection for crop AGB estimation based on data-driven methods. In spite of this, the selected wavebands forming the optimal NDVIs can be interpretable in terms of spectral response of crop biomass. Further studies could focus on the improvement in the representativeness of calibration datasets. It can be realized by collecting wheat samples with different cultivars, growth stages, field treatments, and ecological conditions. Based on the comprehensive datasets, the selected sensitive wavebands might be more stable and the predictive model based these wavebands might have better generalization ability. Another way to improve the stability of narrowband vegetation indices is to develop the vegetation indices, which are specific to a certain crop cultivar and environment condition. Obviously, the latter ones are confined to limited application.

Both PLSR and LSSVM analyses using the continuum removal of hyperspectra between 550 and 750 nm produced better estimation accuracy as compared with the linear regression model based on the optimal NDVI (1193 nm, 1222 nm). The results confirm the capability of red-edge spectra in predicting winter wheat AGB, consistent with previous findings reported by Hansen and Schjoerring [3], Fu et al. [9], and Kanke et al. [38]. They have indicated that the red-edge region contains useful information relevant to crop biomass. It is important to note that the LSSVM regression analysis performed better than the PLSR for winter wheat AGB estimation, when combined with the continuum removal of spectra with different bandwidths. It demonstrated that LSSVM regression analysis was advantageous over PLSR in characterizing the relationships between spectral features and winter wheat AGB during multiple growth stages. In the exploration of the influence of bandwidth on the predictive performance of red-edge spectra, both the LSSVM regression and PLSR models generally got decreased estimation accuracy with increasing bandwidths. However, the continuum removal of spectra with a bandwidth of 1 nm did not yield the highest estimation accuracy. It mainly attributes to the high intercorrelation in neighboring hyperspectral wavebands, which has a negative impact on model accuracy and interpretability. However, PLSR and LSSVM have the ability to partly overcome the multi-collinearity problem. Through increasing the bandwidth, the spectral redundancy

can be reduced. The PLSR and LSSVM regression models achieved the highest estimation accuracy at the bandwidths of 5 and 9 nm, respectively. The obtained estimation accuracies were comparable to those of the PLSR and LSSVM regression models using all spectral wavebands between 400 and 1350 nm (Figure A2 in Appendix A). After the bandwidth exceeded 11 nm, the predictive performance of both PLSR and LSSVM models had a clear decrease (Figure 4a,b). It means that a fraction of spectral features associated with winter wheat AGB are lost when the hyperspectral data are resampled to multispectral data. It also demonstrates the necessity of exploring hyperspectral features for winter wheat AGB estimation. At the bandwidth of 21 nm, the predictive performance of the LSSVM regression model was still acceptable with a RMSE value of 132.05 $g/m^2$. This result is conducive to develop hand-held multispectral devices or low-cost digital cameras for proximal crop monitoring.

*4.2. Performance Comparison of Single-Scale and Multiscale Textures for Winter Wheat AGB Estimation*

With the aim to extract textures that can well characterize the variations of winter wheat canopy during multiple growth stages, a multiscale texture extraction method was proposed (Multiscale_Gabor_GLCM). This method comprehensively utilized the advantages of Gabor filter and GLCM analyses. In the Pearson correlation analyses between winter wheat AGB and textures, the texture metric "Mea-G-S1" extracted by the proposed method exhibited higher correlation with AGB than the best-performing single-scale GLCM-based texture "Var-B" (Figure 5). Yue et al. [18] also reported that the GLCM-based texture metric "Var-B" correlated well with winter wheat AGB. Zheng et al. [17] indicated that the GLCM-based texture metric "Mea" was better than the other seven texture metrics in rice AGB estimation. Through the visualization technique of LSSVM regression, we determined 10 textures (i.e., metric "Mea" from different scales) that contributed most to winter wheat AGB (Figure 7b). In the exploration whether multiscale textures could show superiority to single-scale textures for winter wheat AGB estimation, both PLSR and LSSVM regression models based on the multiscale textures yielded higher $R^2$ and lower RMSE values for the validation dataset compared to the models based on the single-scale textures including GLCM-based textures and textures at each scale (Figures 6–8). The RGB imagery of winter wheat canopy is usually composed of soils, leaves, stems, ears, and shades resulted from leaf clumping. The proportion of each component changes and wheat canopy structure gradually becomes complicated during growing seasons. At the jointing stage, wheat canopy cannot fully cover soil background, and soils can be seen from imagery. From the heading stage to anthesis stage, wheat ears and flowers emerge, and wheat canopy almost fully covers soil background. From the anthesis stage to early grain filling stage, wheat ears become plump and leaves begin to turn yellow. These wheat canopy dynamic changes reflected in digital images can be captured by fine image textures. Moreover, winter canopies usually exhibit heterogeneity within or among fields due to different cultivars, field treatments, and cropland micro-meteorology, even at the same growth stage. Therefore, using single-scale textures will hardly capture the spatial changes of wheat canopies during growing seasons. This is confirmed by the predictive performance comparison between multiscale and single-scale textures for winter wheat AGB estimation. The proposed multiscale texture extraction method firstly utilized Gabor filters with multiple scales and orientations to generate textural images (i.e., Gabor magnitude images). These Gabor textures are less sensitive to variations of local lighting conditions, which will improve the reliability of subsequent image processing. Then GLCM analyses were conducted on these images to extract finer textures. Our previous study [20] has found that the textures of different orientations did not show significant heterogeneity for winter wheat canopy imagery; thus, this study only focused on the scale properties of textures derived from UAV-based RGB imagery. The optimal scale of textures is highly correlated with crop planting density and canopy size. However, the quantitative relationships among them are still unclear. To obtain abundant discriminative features, we followed the previous research results [20,32] and set five different scales,

which determined the spatial frequencies of Gabor filters. In the study of Yue et al. [18], they resampled original UAV-based RGB images to images with spatial resolutions of 2, 5, 10, 15, 20, 25, and 30 cm, respectively. Then they extracted GLCM-based textures from these images for winter wheat AGB estimation. The aim of doing this was also to extract multiscale textures to represent winter wheat canopy variations during multiple growth stages. They got improved estimation accuracy of winter wheat AGB by combining PLSR and the GLCM-based textures derived from RGB images with different spatial resolutions, which was consistent with the findings in our study. However, it seemed a little arbitrary in the determination of spatial resolution of images for analysis. In our proposed multiscale texture extraction method, the scale of textures was controlled by the scale parameter of Gabor filters. Further studies should investigate the quantitative relationships between crop canopy structure and scale parameters. It could help to determine the optimal scale parameter when extracting multiscale textures for crops of interest. In this study, the proposed multiscale texture extraction framework can also involve other multiresolution image analysis methods such as 2D wavelet transformation and 3D Gabor filter.

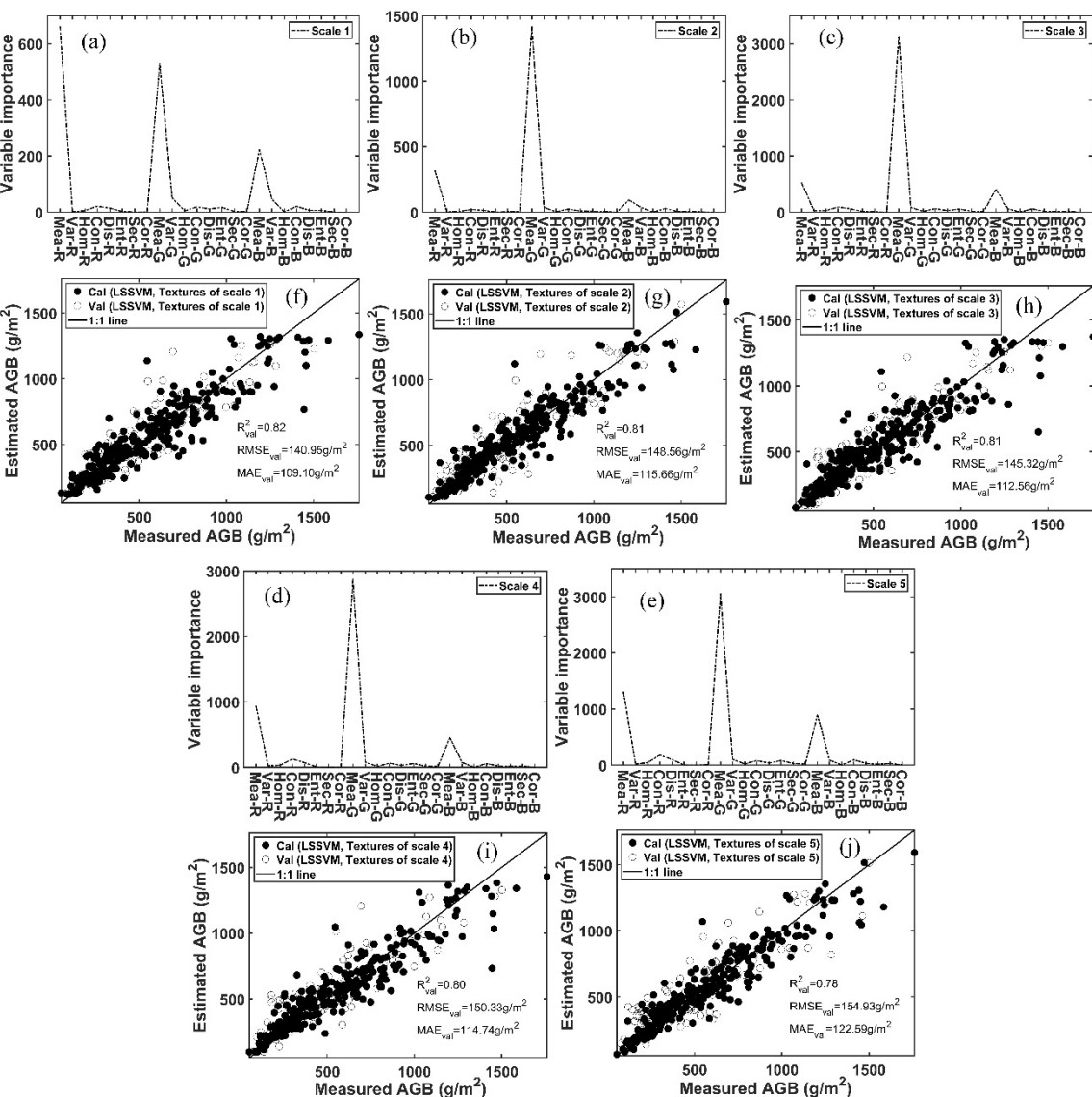

**Figure 8.** Importance values (**a**–**e**) and predictive performance (**f**–**j**) of the textures at each scale based on LSSVM regession analyses.

*4.3. Evaluation of the Combined Use of Hyperspectral and Textural Features for Winter Wheat AGB Estimation*

Before the combined use of hyperspectral and textural features, the predictive performances of them were compared for winter wheat AGB estimation. Compared with the models based on the continuum removal of red-edge spectra (SpeCR, FWHM = 9), the models based on the multiscale textures had comparable estimation accuracy. It confirms the utility of textures derived from high-definition RGB images for winter wheat biomass estimation, since UAV-based RGB images are more readily acquired than ground-based, airborne, and spaceborne hyperspectral data. In the exploration of whether LSSVM analyses conducted using the combination of optimal NDVI, SpeCR (FWHM = 9), and multiscale textures could further improve estimation accuracy of winter wheat AGB, the LSSVM regression model based on the combination of the SpeCR (FWHM = 9) and selected multiscale features yielded the higher $R^2$ and lower RMSE for validation dataset, compared to the other combinations (i.e., multiscale textures + optimal NDVI and multiscale textures + optimal NDVI + SpeCR). However, the combination did not significantly improve the estimation accuracy compared to the use of SpeCR or multiscale textures only. The results are not in agreement with the findings reported by Yue et al. [18] and Zheng et al. [17], who highlighted that the combination of spectral and textural features could improve the estimation accuracy of crop AGB. The primary reason for this is that the multiscale textures in the study perform better than the GLCM-based textures used in [17,18]. The addition of red-edge hyperspectral information could not make an improvement in estimation accuracy. However, the spectral features used in [17,18] were broadband vegetation indices or some common narrowband vegetation indices, such as NDVI (670, 800 nm), optimized soil adjusted vegetation index (OSAVI (670, 800 nm)), and two-band enhanced vegetation index (EVI2), which did not utilize the informative information from red-edge spectra well.

## 5. Conclusions

This study investigated the utility of multiscale textures extracted from high-definition UAV-based RGB images in predicting winter wheat biomass during multiple growth stages. The proposed multiscale texture extraction method (Multiscale_Gabor_GLCM) took good advantage of Gabor filters and GLCM analyses. The feature selection function embedded in LSSVM regression helped identify 10 influential multiscale textures to winter wheat biomass. Both the PLSR and LSSVM regression models using the multiscale textures produced better estimation accuracies than those using single-scale GLCM-based textures. Compared to the spectral features based models, the multiscale textures got comparable estimation accuracy to the SpeCR with a bandwidth of 9 nm. Although the selected best-performing NDVI (1193, 1222 nm) got intermediate estimation accuracy, it had great value in practical application due to its ease of use and low computation. The LSSVM regression analyses conducted using the combination of multiscale textures and hyperspectral features did not significantly improve the estimation accuracy for winter wheat AGB, compared to the use of multiscale textures or SpeCR alone. The LSSVM regression and its visualization technique showed great potential in establishing a high-accuracy winter wheat estimation model and understanding the driving force behind the model. This study confirmed the reliability of multiscale textures, red-edge spectra, and optimal NDVI calculated by water-sensitive wavebands for winter wheat AGB estimation. Further studies are required to verify the effectiveness of multiscale texture extraction method in predicting crop biomass at different ecological regions, especially for the high-yielding crops.

**Author Contributions:** Conceptualization: Y.F.; methodology: Y.F.; software: Y.F.; validation: Y.F.; formal analysis: Y.F.; investigation: Y.F.; resources: G.Y. and H.F.; data curation: G.Y., X.X., H.F., and X.S.; writing—original draft preparation: Y.F.; writing—review and editing: Y.F. and G.Y.; visualization: Y.F.; supervision: G.Y., Z.L., and C.Z.; project administration: G.Y. and C.Z.; funding acquisition: Y.F, G.Y., and C.Z. All authors have read and agreed to the published version of the manuscript.

**Funding:** This study was supported by the Natural Science Foundation of China (41801225), the Key-Area Research and Development Program of Guangdong Province (2019B020214002, 2019B020216001), the National Key Research and Development Program of China (2017YFE0122500) and the Beijing Million Talent Project (No.2019A10).

**Institutional Review Board Statement:** Not applicable.

**Informed Consent Statement:** Not applicable.

**Data Availability Statement:** The data presented in this study are available on request from the corresponding author.

**Acknowledgments:** The authors extend gratitude to Bo Xu, Weiguo Li, Lin Wang and Hong Chang for their assistance in field data collection. The detailed comments from the editors and reviewers improved our study.

**Conflicts of Interest:** The authors declare no conflict of interest.

## Appendix A

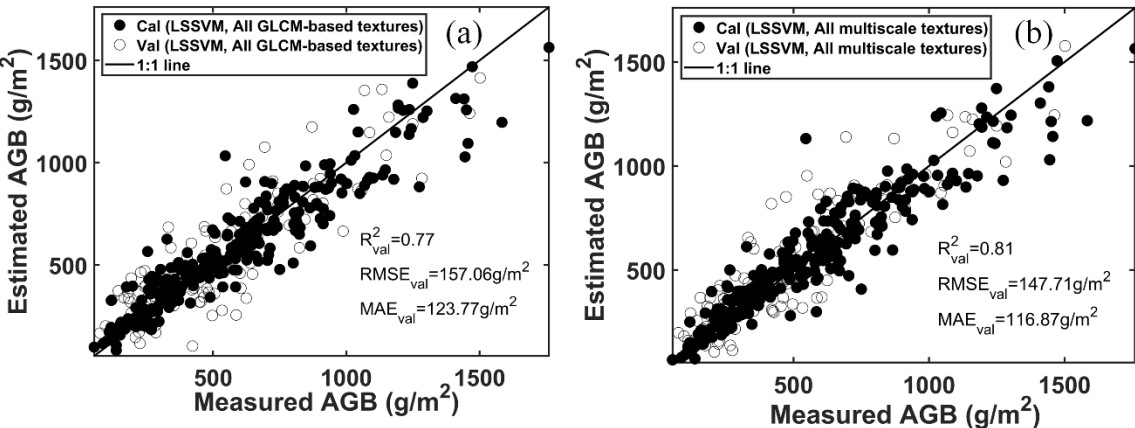

**Figure A1.** Predictive performance of the LSSVM regression models based on (**a**) all GLCM-based textures and (**b**) all multiscale textures for winter wheat AGB estimation.

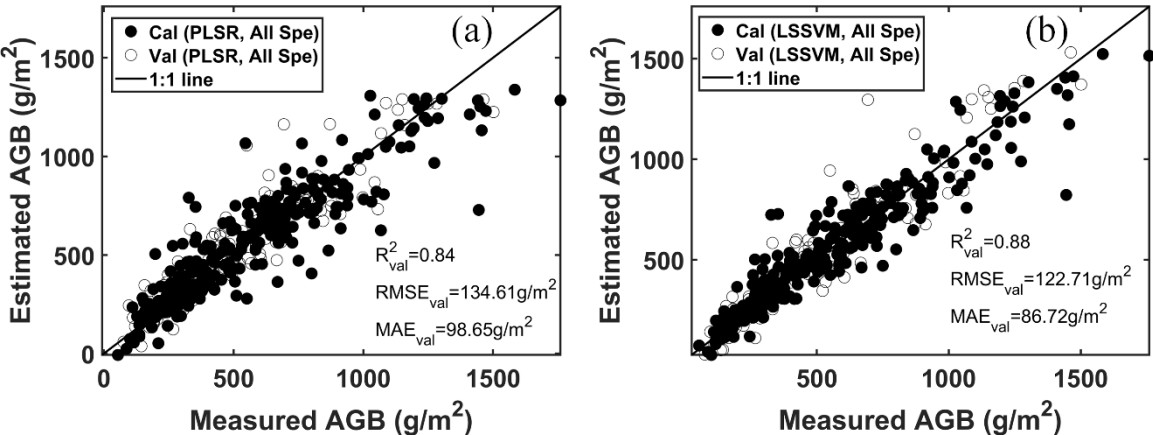

**Figure A2.** Predictive performance of the PLSR model (**a**) and LSSVM regression model (**b**) using all spectra (400~1350 nm) for winter wheat AGB estimation.

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
