# Peer review of "Improved Estimation of Winter Wheat Aboveground Biomass Using Multiscale Textures Extracted from UAV-Based Digital Images and Hyperspectral Feature Analysis"

_remotesensing, doi:10.3390/rs13040581_

Round 1

Reviewer 1 Report

The article has very dense information about the use of spectral data for wheat biomass estimation. The content is clear, well developed and the experimental proceedings are well designed. The statistics used are also robust and several approaches are compared, with several important considerations being made by the authors about the advantages and disadvantages of each approach. 

The conclusions are perfectly supported by the obtained data. 

There are only two modifications that can be suggested to improve the text. 

The authors must make some efforts to shorten the abstract. There are things that can be condensed in shorter phrases without loosing sense. Some things are well developed in the article text and do not need to be so detailed in the abstract. 

In the materials and methods (line 189/190), the location coordinates are given in “dd mm ss” format. Probably nowadays, with the popularization of online mapping systems, it’s more practical to use the “dd.ddd” presentation. The coordinate system should also be provided (geographic/WGS84; EPSG 4326?). I was able to locate the region with an online map service but only after some edition of the coordinates syntax. 

Apart of the article, it would be interesting if the authors could go further and, considering the imbricated workflow, could provide an automation routine for this kind of analysis, for instance as a Quantum GIS plugin. 

Reviewer 2 Report

I can only recommend the manuscript for a major revision at this stage. Please see my comments below.

The abstract is extended needs to be shortened.

Texture matrices from RGB images or hyperspectral images? This should be clarified from the beginning.

What are the differences between the multiscale and single scale GLCM-based textures?

Line 42 and 47, “got decreased … ”, “got comparable estimation accuracy …” such informal language use not appropriate.

Keywords are too long. Improve!

Lines 69-72, these statements need to be backed up with relevant studies.

Line 78, should be LiDAR.

Line 136, define ultra-high spatial resolution images.

Line 149, Gabor filters needs to be expanded a little bit.

Lines 171 and 172, what are those studies?

Line 188, why the 2015-2016 growing season skipped?

Line 215, provide detailed camera information. Same on line 218.

Line 237, what was the size of your ROR? How did you determine the location of the plot from the images? What were image registration accuracy from the different time points?

Until I read the line 241, I thought you used hyperspectral imagery in this study. This should be from the beginning like in the title and the abstract.

Line 245, what was the size of the sensor footprint?

Line 271, what are those five scales?

Line 278, how did you come up with these 15 magnitude images?

Still, it is not clear why you called multiscale?

Line 305, is that the correct red-edge regions? Confirm!

The authors already tested the performance of different data types, regression methods, etc. as a logical follow-up, It would be nice to evaluate the performance of the best approach by grouping the in terms of growth stages and cultivars, respectively. This would help to understand the generalization ability of the best model and proposed method.

Reviewer 3 Report

Abstract - need to be more clear and shorter than it is. It is recommended ti integrate only the most relevant results as well as the robustness of the approach and not a lot of technical details.

State-of-the-art is useful but it could be improved. The presentation is detailed and there are only few distinct paragraphs. It might be better to follow the methodological direction and the sensor employment direction in text re-shaping (only re-organize a little the existing text).

Please check if each of the methods/techniques have 1-2 relevant references (usually they have). 

Objectives (rows 176-185) needs a higher visibility (a distinct paragraph).

Materials and methods

There is no map of the location of study area in China...

Soil features can be provided in a table. Some photos from study area can be interesting for understanding the starting point of your approach.

Row 197 - experiments is correct.

The approach is interesting and have important potential practical use. It could be fine to find a solution in integrating a flow chart with the terrain experiment stages together with the remote sensing image processing stages (figure 2 is a step forward). Figure 1 is good but it is provided a little abrupt in the structure of the approach. 

It is essential to open your research results to agricultural specialists. It is better to provide some images from the remote sensing field data collection with the instruments used and the experiment plot samples. This can give more efficient information to the potential users of your methods.

Fig. 1 - it is useful to provide some detail samples from processed imagery in order to better adapt the approach to the scope of the journal. 

Section 2.4 - it is recommended to explain their significance for data mining based on the available field collected experimental data (examples).

Row 340 - adopted is correct.

Figure 2 is essential in understanding your approach. It can integrate in the same a validation stage of the results. It can be explained a little more.

Results.

There is a big amount of statistical data. Sometimes it can be provided using synthetic tables (table 3 is a good example) and indicate only the essential parameters in order to explain the significance of the current approach (already done).

Some processed imagery samples with point sample position can help better the understanding of the results.

Discussion 

Section 4.1 - the interpretation is good but it can be resumed in a complex spectral diagram showing the curves of the investigated crop and the characteristic points along them with significance in biomass estimation using the red edge and other features. The influence of soil spectral features can be also explained in the diagram.

Conclusion - it might be useful to put the emphasis on some practical significance of the results and if the approach can be adapted to other data for larger areas like Sentinel 2 MSI images which also contain a red-edge band.

It can be also interesting to explain the relationship between the spatial resolution of data sampling and the accuracy of the biomass models obtained.

Round 2

Reviewer 2 Report

I am satisfied with the authors' responses. The only minor change I'd like to see is the incorporation the point 13 in the response letter. Please briefly and concisely include the response in the manuscript. 
